# Effect of a Low-Methane Diet on Performance and Microbiome in Lactating Dairy Cows Accounting for Individual Pre-Trial Methane Emissions

**DOI:** 10.3390/ani11092597

**Published:** 2021-09-03

**Authors:** Juana C. Chagas, Mohammad Ramin, Ruth Gomez Exposito, Hauke Smidt, Sophie J. Krizsan

**Affiliations:** 1Department of Agricultural Research for Northern Sweden, Swedish University of Agricultural Sciences (SLU), Skogsmarksgränd, 90183 Umeå, Sweden; mohammad.ramin@slu.se; 2Laboratory of Microbiology, Wageningen University & Research, 6708 WE Wageningen, The Netherlands; ruth.gomezexposito@wur.nl (R.G.E.); hauke.smidt@wur.nl (H.S.)

**Keywords:** dairy cow, enteric methane, feed efficiency, grass silage, maize silage, rapeseed oil, rumen microbiome

## Abstract

**Simple Summary:**

Low methane-emitting dietary ingredients have been identified in extensive research conducted during the past decade. This study investigated the effects of replacing grass silage with maize silage, with or without rapeseed oil supplementation, on the methane emissions and performance of dairy cows. Pre-trial measurements of methane-emissions were used in the evaluation. Partial replacement of grass silage with maize silage did not affect methane emissions but reduced dairy cow performance. Adding rapeseed oil to the diet substantially reduced methane emissions due to modified rumen microbiota, resulting in impaired nutrient intake, digestibility, and yield of energy-corrected milk. Correcting for individual cow characteristics of methane emissions did not affect the magnitude of suppression of methane emissions by dietary treatments.

**Abstract:**

This study examined the effects of partly replacing grass silage (GS) with maize silage (MS), with or without rapeseed oil (RSO) supplementation, on methane (CH_4_) emissions, production performance, and rumen microbiome in the diets of lactating dairy cows. The effect of individual pre-trial CH_4_-emitting characteristics on dietary emissions mitigation was also examined. Twenty Nordic Red cows at 71 ± 37.2 (mean ± SD) days in milk were assigned to a replicated 4 × 4 Latin square design with four dietary treatments (GS, GS supplemented with RSO, GS plus MS, GS plus MS supplemented with RSO) applied in a 2 × 2 factorial arrangement. Partial replacement of GS with MS decreased the intake of dry matter (DM) and nutrients, milk production, yield of milk components, and general nutrient digestibility. Supplementation with RSO decreased the intake of DM and nutrients, energy-corrected milk yield, composition and yield of milk fat and protein, and general digestibility of nutrients, except for crude protein. Individual cow pre-trial measurements of CH_4_-emitting characteristics had a significant influence on gas emissions but did not alter the magnitude of CH_4_ emissions. Dietary RSO decreased daily CH_4_, yield, and intensity. It also increased the relative abundance of rumen Methanosphaera and Succinivibrionaceae and decreased that of Bifidobacteriaceae. There were no effects of dietary MS on CH_4_ emissions in this study, but supplementation with 41 g RSO/kg of DM reduced daily CH_4_ emissions from lactating dairy cows by 22.5%.

## 1. Introduction

The methane (CH_4_) concentration in the Earth’s atmosphere has been rising rapidly over the past decade and is affecting the climate. Data suggest that the increase in global CH_4_ emissions recorded from 2005 to 2015 is due to the increased extraction of shale gas, and that natural gas and oil industries are the main contributors, rather than agriculture [1]. However, CH_4_ emissions from the agricultural sector comprise 43% of total non-CO_2_ greenhouse gas (GHG) emissions [2], representing 25% or 3.5 Gt CO_2_-eq of total global anthropogenic emissions [3]. Population growth and rising incomes in developing countries are leading to increasing demand for animal products. However, the emissions of non-CO_2_ GHG in Europe are expected to decrease by 1.5% by 2030 compared to 2008 [4], and mitigating emissions to limit global warming to 1.5 °C by 2100 will demand much more effort.

Several dietary strategies to reduce CH_4_ emissions have been investigated over the years, including strategies for dairy production in northern Europe, which is characterised by grass silage-based feeding. In general, total CH_4_ emissions increase with earlier harvest of grass for silage production, which can be attributed to higher dry matter intake (DMI) and a more digestible diet [5]. Replacing grass silage (GS) with maize silage (MS) has been suggested to promote increased propionate rather than acetate fermentation in the rumen, and thereby decrease CH_4_ production in dairy cows [6]. Van Gastelen et al. [7] observed a decrease in CH_4_ emissions of between 8% and 11% when MS completely replaced GS in the diet of dairy cows. Another suggested dietary alternative for efficiently reducing enteric CH_4_ emissions is the inclusion of rapeseed oil (RSO) in the diet of dairy cows, e.g., Bayat et al. [8] and Villar et al. [9] obtained reductions of up to 23% in CH_4_ emissions with inclusion of 5% RSO in the diet of lactating dairy cows.

Thus, it appears that nutrition and feeding approaches may be able to reduce CH_4_ emissions. According to Knapp et al. [10], the dietary mitigation effect on CH_4_ emissions per unit of energy-corrected milk (ECM) varies between 3% and 15%, but greater long-term reductions, of up to 30%, can be achieved by combined genetic and feeding management approaches.

Improved feed efficiency through targeted breeding [11] and improved longevity or lifetime productivity [12] have been suggested as the best long-term strategies to reduce CH_4_ emissions from dairy cows. Studies have revealed differences in the CH_4_-emitting phenotype of ruminants [13,14,15]. Thus, traits such as CH_4_ emissions yield and intensity [16], as well as residual CH_4_ emissions (observed minus predicted CH_4_ production), have been suggested in order to select for lower-emitting dairy cows [17,18]. However, it is not known whether the effect of dietary CH_4_ emission mitigation strategies differs between low- and high-emitting animals.

The hypotheses tested in this study were that dairy cows fed a diet containing MS would produce less CH_4_ than if only fed GS; that moderate supplementation with RSO would further mitigate CH_4_ emissions, without any negative impact on feed intake, digestion, and milk production; and that the dietary CH_4_ emission-mitigating effect is lower than previously reported when related to observed differences in CH_4_-emitting phenotype of the cows. Specific objectives of the study were to investigate the effects of RSO supplementation on dairy cow CH_4_ emission traits, performance and microbiota composition when partly replacing GS with MS in the diet of lactating dairy cows, and to establish the effect of individual cow CH_4_-emitting characteristics on dietary CH_4_ emissions mitigation.

## 2. Materials and Methods

A feeding trial was conducted at Röbäcksdalen experimental farm of the Swedish University of Agricultural Sciences in Umeå (63°45′ N, 20°17′ E) from January to May 2019. Handling of animals in the trial was approved by the Swedish Ethics Committee on Animal Research (Dnr. A17/2016 + A33/16), represented by the Court of Appeal for Northern Norrland in Umeå, and the experiment was carried out in accordance with laws and regulations governing experiments performed with live animals in Sweden.

### 2.1. Cows and Pre-Trial Measurements

Prior to the experiment, a pre-trial of seven days was carried out [19]. Twenty Nordic Red Swedish dairy cows (12 multiparous and eight primiparous) weighing 601 ± 81.9 (mean ± SD) kg, at 71 ± 37.3 days in milk and producing 34.2 ± 5.26 kg of milk/d at the beginning of the experiment were monitored for CH_4_-emitting characteristics. During the pre-trial, all cows were fed the same total mixed ration (TMR) consisting of 754:179:62:4 grass silage: crimped barley: heat-treated rapeseed meal (ExPro-00SF; AarhusKarlshamn Ltd., Malmö, Sweden): minerals (Mixa Optimal; Lantmännen Lantbruk AB, Malmö, Sweden) in g/kg diet DM. The cows were monitored for DM intake (DMI) and CH_4_ production, and their body weight (BW) was recorded after morning milking on a minimum of two days in the beginning of the week and two days at the end of the week. These data were used to estimate the pre-trial CH_4_-emitting value of each cow.

### 2.2. Housing, Experimental Design and Diets

The cows were housed in an insulated free-stall barn equipped with an automatic feed intake recording system and fresh water sources. The cows were fed a TMR ad libitum four times per day, at 0300, 0800, 1400, and 1800 h, using an automatic feeding wagon, and were milked twice a day, at 0600 and 1630 h.

The cows were blocked by parity and milk yield (MY) and the experimental treatments were randomly assigned to all cows within each of the five blocks. The experiment was conducted as a replicated 4 × 4 Latin square design and with four experimental periods lasting 28 d each with a total experiment duration of 112 d. All recordings and samplings were performed during the last 14 d of each experimental period.

Dietary treatments were applied in a 2 × 2 factorial arrangement and consisted of: GS, GS supplemented with RSO (GSO), GS plus MS (GSMS), and GSMS supplemented with RSO (GSMSO). The diets without RSO, i.e., GS and GSMS, were composed of: GS (539 and 270 g/kg dry matter; DM), MS (0 and 259 g/kg DM), crimped barley (353 and 352 g/kg DM), heat-treated rapeseed meal (RSM; 93 and 93 g/kg DM) (ExPro-00SF; AarhusKarlshamn Ltd., Malmö, Sweden), and mineral and vitamin mix feed (MM; 15 and 15 g/kg DM) (Mixa Optimal; Lantmännen Lantbruk AB, Malmö, Sweden), respectively. The diets supplemented with RSO, i.e., GSO and GSMSO, were composed of: GS (519 and 259 g/kg DM), MS (0 and 259 g/kg DM), crimped barley (333 and 334 g/kg DM), RSO (40 and 40 g/kg DM), RSM (93 and 93 g/kg DM), and MM (15 and 15 g/kg DM), respectively.

The first cut of GS, primarily timothy grass (*Phleum pratense*), but containing some (seed ratio 80:20; botanical analysis not made) red clover (*Trifolium pratense*) was harvested in Umeå on 8 to 9 June 2018. The silage was preserved using a formic acid-based additive (PromyrTM XR 630, Perstorp, Sweden; 3.5 L/t) and stored in a bunker silo. The maize (*Zea mays* L.) silage was purchased from Denmark and was from a harvest from 2017. The maize silage was stored in a bunker silo and baled in 2018 before transportation to Umeå. The barley (*Hordeum vulgare*) was harvested in Umeå on 17 August 2018, treated with 3.5 L/t of propionic acid and stored as crimped barley in air-tight bags (1.6 m × 60 m, Ltd. Rani Plast Oy, Terjärv, Finland). The RSO product used was manufactured by AAK Sweden AB (Karlshamn, Sweden) and had a concentration of polysaturated fatty acids of 280 g/kg of DM and a metabolisable energy (ME) content of 32.5 MJ/kg of DM (AAK, Sweden).

The chemical composition and nutritional value of the dietary ingredients are shown in Table 1. The dietary ingredients were mixed just before each feeding, using a TMR mixer (Nolan A/S, Viborg, Denmark).

### 2.3. Data Recording and Sampling

Individual feed intake was recorded daily by Roughage Intake Control feeders (Insentec B. V., Marknesse, The Netherlands) and daily MY was recorded using a gravimetric milk recorder (SAC, S.A. Christensen and Co Ltd., Kolding, Denmark). Feed intake and MY are reported only for d 15–28 of each period. The BW of the cows was recorded at the beginning of the study and then every week after morning milking.

Mass fluxes of CH_4_, carbon dioxide (CO_2_), and oxygen (O_2_) were recorded daily using an open-circuit head chamber system (GreenFeed system, C-Lock Inc., Rapid City, SD, USA) as described by [19]. Gas calibrations were performed once a week, and CO_2_ recovery tests were conducted every second week, on d 14, in each experimental period. The air filters were cleaned twice a week in order to maintain the airflow above 26 L/s. Concentrate pellets (GFC; Komplett Amin 220; Lantmännen Lantbruk AB, Malmö, Sweden) were provided to the cows in the GreenFeed unit to ensure regular visits (i.e., an average of five visits per day) and capture gas emissions over a 24-h cycle. The GreenFeed unit was operated continuously during the experiment, but gas data are reported only for d 15–28 of each period.

Milk samples were collected twice a day, at 0600 and 1630 h, from d 19 to 21 and from d 26 to 28 of each period. The samples were stored in plastic bottles with the preservative Bronopol (bottles provided by Valio Ltd. (Helsinki, Finland) at 5 °C until analysis. The diets were adjusted weekly according to changes in DM concentration (oven-dried at 60 °C for 48 h) for the silages and the concentrate feeds. The dried samples were milled (SM 2000; Retsch Ltd., Haan, Germany) to pass through a 2- or 1-mm sieve, depending on analytical purposes. Silage samples were taken once per week and stored at −20 °C for the analysis of fermentation quality. The frozen silages were milled to pass through a 20-mm sieve and kept frozen until analysis.

Rumen fluid was collected from all cows used in the experiment, on one occasion per experimental period from d 19 to 21. On each sampling day, each cow was restrained after the morning milking and rumen fluid samples were collected using a stomach tube (RUMINATOR; Munich, Germany) as described by [22]. The first sample of rumen fluid, comprising about 500 mL, was discarded, in order to avoid saliva contamination. Then, a sample of 500 mL was taken and filtered through a two-layer cheesecloth. Subsamples for microbial analysis were transferred to 2.0 mL Eppendorf tubes, immediately frozen on dry ice, and kept at −80 °C in a freezer until analysis.

Apparent diet digestibility was assessed by collecting faecal samples (300 mL) from the rectum of all experimental cows twice a day, at 0900 and 1500 h, from d 22 to 24 in each experimental period. Composite faecal samples per cow and period were obtained at the end of each sampling period. The samples were oven-dried at 60 °C for 48 h and then milled to pass through a 1-mm sieve in a cutter mill. Faecal samples used for indigestible NDF (iNDF) analysis were ground by mortar and pestle to pass through a 2.5-mm sieve.

### 2.4. Chemical Analysis

The DM of feed ingredients and faeces was determined by oven drying at 105 °C for 16 h, and ash content was determined by combustion of the dried samples at 500 °C for 4 h (AOAC, 2012; method 942.05). Organic matter (OM) was determined as 1000-ash. Oven DM concentration for the silages was corrected for volatile losses according to [23]. Total nitrogen (N) in the samples was analysed with the Kjeldahl method [24] (method 990.03) using a Heating Block (SEAL Analytical, Mequon, WI, USA) and an AutoAnalyzer 3 Unit (SEAL Analytical, Mequon, WI, USA). Crude protein (CP) concentration was calculated as N × 6.25. Neutral detergent fibre (NDF) content, reported as ash-free, was analysed according to Van Soest et al. [25] with heat-stable α-amylase and sodium sulphite [26], using the filter bag technique in an Ankom200 digestion unit (Ankom Technology Corp., Macedon, NY, USA). The concentration of NDF was determined on an ash-free basis by combustion of residual material in the Ankom bags at 500 °C for 4 h.

To determine iNDF concentration in the feed ingredients and faeces, 2 g (±0.1) samples were weighed into polyester bags of 11 µm pore size. These were subjected to 288 h of in situ incubation [27] in three rumen-cannulated lactating cows fed a TMR based on grass silage (600 g/kg of DM) and commercial concentrate (400 g/kg of DM), as described by Krizsan et al. [28]. The iNDF concentration was expressed as ash-free. The starch content in MS was determined by the amyloglucosidase method according to Salo and Salmi [29], using an UV-VIS spectrophotometer (UV-1800; Schimadzu Co., Kyoto, Japan). Crimped barley [30], RSM [20], and GFC starch concentration (reported by Lantmännen Lantbruk AB, Sweden) were determined by near infrared spectroscopy (NIRS). Crude fat concentration in GS and MS (reported by Eurofins Agro Testing AB, Sweden), in RSM and GFC (reported by Lantmännen Lantbruk AB, Sweden), and in crimped barley [30] were determined by NIRS.

The frozen silage samples were thawed and pressed, and pH in the press liquid was measured with a pH meter (Metrohm, Herisa, Switzerland). Ammonium nitrogen (NH_3_-N) was analysed according to Broderick and Kang [31], by direct distillation after adding MgO in a Kjeltec 2100 Distillation Unit (Foss Analytical Ltd., Hillerød, Denmark). The concentrations of volatile fatty acids (VFA) and lactic acid were analysed according to Ericson and André [32].

The milk samples were analysed for fat, protein, urea, and lactose concentration at the laboratory of Valio Oy (Seinäjoki, Finland) using infrared reflectance spectroscopy (MilkoScan TM FT120, Foss Electric, Hillerød, Denmark).

### 2.5. Microbial Analysis

#### 2.5.1. DNA Isolation

Rumen fluid (500 µL) was centrifuged for 10 min at 21,000× *g* and 4 °C and the supernatant was discarded. The pellet was re-suspended in 700 µL of stool transport and recovery (STAR) buffer (Roche Diagnostics Nederland BV, Almere, The Netherlands) and transferred to a screw cap tube containing 0.25 g of 0.1 mm glass beads. The samples were subjected to repeated bead beating (5.5 m/s × 3 × 60 s), followed by 15 min heating at 95 °C and 1000 rpm on a rotary shaker and centrifugation for 5 min at 4 °C at 21,000× *g*. The supernatant was transferred to a separate tube and maintained cold. The pellet was subjected to a second round of cell lysis with another 300 µL of STAR buffer. Supernatants from both cycles were pooled and 250 µL were used for DNA isolation using the Maxwell 16 Tissue LEV Total RNA Purification Kit (Promega, Madison, WI, USA). DNA was eluted in 50 µL nuclease-free water. Negative controls, using only reagents and no sample, were also included. The quantity of DNA was fluorometrically determined using Qubit in combination with the dsDNA BR Assay Kit (Invitrogen, Carlsbad, CA, USA) following the manufacturer’s recommendations. The DNA was diluted to ~20 ng/µL and stored at −20 °C until further use.

#### 2.5.2. 16S rRNA Gene Amplicon Sequencing

Microbiota composition was analysed with barcoded amplicons of the V4 region of the 16S rRNA gene generated using the F515-806R primer set [33]. The amplification reactions were performed in triplicate as described elsewhere [34]. After confirmation of the correct size of the amplicons by agarose gel electrophoresis, PCR products were purified with the HighPrep kit (MagBioEurope Ltd., Kent, UK) following the manufacturer’s instructions. The PCR products were pooled in equimolar amounts and sequenced on the Illumina NovaSeq 6000 platform (GATC-Biotech, Konstanz, Germany). To control for potential technical biases, two human gut mock synthetic communities [35] and three rumen mock synthetic mock communities were included as positive controls, and PCR reactions with no DNA template as negative controls.

#### 2.5.3. qPCR of Ciliate Protozoa

Absolute quantification of the 18S rRNA genes from ciliate protozoa was performed using primers Syl316f and Syl539r, following the amplification conditions described by Sylvester et al. [36]. In brief, all reactions were performed in triplicate in volumes of 10.5 µL, using 5 µL of iQ SYBR Green Supermix (Bio-Rad Laboratories B.V.), 2.6 µL nuclease-free water, 200 mM (final concentration) of each primer, and 2.5 µL of either the DNA template (~1 ng/µL) or nuclease-free water, using a CFX384 Real-Time PCR Detection System (Bio-Rad Laboratories, Veenendaal, The Netherlands). The amplification conditions consisted of an initial denaturalisation at 94 °C for 4 min, followed by 45 cycles of 94 °C for 30 s, 54 °C for 30 s, 72 °C for 60 s, and a final elongation at 72 °C for 6 min. For each assay, fluorescence was detected at the end of each cycle and the specificity of the PCR reactions was determined by including melting curves resulting from increasing the temperature from 60 to 95 °C in increments of 0.5 °C. Standard curves (10^1^–10^8^ copies/µL) were prepared from the 18S rRNA gene obtained from *Epidinium caudatum.*

### 2.6. Calculations

The CH_4_-emitting characteristic of the individual cows was calculated from average measured CH_4_ minus predicted CH_4_. The predicted CH_4_ was determined from bi-variate regression of DMI and BW on measured CH_4_ production. The chemical composition of diets was calculated based on the intake, dietary ingredient composition determined from fresh weight proportions and ingredient chemical composition. Total intake was calculated as TMR intake plus GFC intake. The apparent digestibility of nutrients was calculated using iNDF as an internal marker in feeds and faeces [27]. Potentially digestible NDF (pdNDF) was calculated as NDF—iNDF. The ME content and protein balance in the rumen (PVB) were calculated based on coefficients from feed tables [20]. Metabolisable protein (MP) was calculated according to Spörndly [21]. Milk constituent concentrations were calculated as a weighted mean of the combined morning and afternoon milk yields. Daily ECM yield was calculated according to Sjaunja et al. [37]. Feed efficiency was calculated as daily yield of ECM/daily amount of DMI, and milk N efficiency (MNE) as the ratio of N milk yield in grams to N intake in kilos. Respiratory quotient (RQ) was calculated as the ratio between CO_2_ eliminated and O_2_ consumed on a molar basis [38].

Raw 16S rRNA gene sequences data for archaea and bacteria were processed in the NG-Tax 2.0 pipeline [39], using the default settings and the SILVA 132 SSU reference database [35]. Read counts were normalised to relative abundance and compositional analysis was performed in R version 3.5.0, using the packages phyloseq (v1.24.2), ape (v5.3), microbiome (v1.2.1) and ggplot2 (v3.3.2). The raw sequence data generated for this study can be found in the European Nucleotide Archive (ENA) under accession number PRJEB43834. The relative abundance data were used as input for variance analysis. Finally, the average copy number of ciliate protozoa per sample was calculated per mL of rumen fluid, and the values were used as input for variance analysis (sequence data in ENA under accession number AM158474.1).

### 2.7. Statistical Analysis

Experimental data (except for gas emissions) were subjected to analysis of variance using the MIXED procedure in SAS (SAS Inc. 2002–2003, Release 9.4 SAS Inst., Inc., Cary, NC, USA) by applying the following model:Y_ijkl_ = µ + B_i_ + P_j_ + C_k_(B)_i_ + D_l_ + ε_ijkl_(1)
where Y_ijkl_ is the dependent variable, µ is the mean of all observations, B_i_ is the fixed effect of block i, P_j_ is the fixed effect of period j, C_k_ (B)_i_ is the random effect of cow k within block i, Dl is the fixed effect of diet l, and ε_ijkl_ is the normally distributed random residual error with an expected mean of zero and constant variance.

Gas emissions data were subjected to analysis of variance using the MIXED procedure in SAS, and with residual CH_4_ as covariate, according to the model:Y_ijkl_ = µ + β(X_ijkl_ − X) + B_i_ + P_j_ + C_k_(B)_i_ + D_l_ + ε_ijkl_(2)
where Y_ijkl_ is the dependent variable, µ is the mean of all observations, β (X_ijkl_ −X) is the fixed effect of covariate, B_i_ is the fixed effect of block i, P_j_ is the fixed effect of period j, C_k_ (B)_i_ is the random effect of cow k within block i, D_l_ is the fixed effect of diet l, and ε_ijkl_ is the normally distributed random residual error with an expected mean of zero and constant variance.

Least square means are reported for all parameters evaluated. Mean separation and the 2-way interaction between forage and oil were investigated by orthogonal contrasts. Differences were considered significant at *p* ≤ 0.05.

## 3. Results

### 3.1. Pre-Trial Measurements of Intake, Body Weight and CH_4_ Emissions

Pre-trial DMI, BW, and CH_4_ production for all experimental cows was 21.3 ± 2.87 (mean ± SD) kg/d, 609 ± 92.6 kg, and 382 ± 83.0 g/d, respectively. Residuals based on observed minus predicted values of CH_4_ regressed on predicted CH_4_ emissions are shown in Figure 1.

### 3.2. Experimental Dietary Ingredients and Diets

The GS and MS diets were comparable in terms of ME concentration, despite differences in CP and iNDF, reflecting the different chemical energy sources of the forages (Table 1). Further, the GS and MS were both well-fermented, with a low pH and relatively low concentration of fermentation acids. The GS displayed more extensive lactic acid fermentation and had a lower NH_3_-N concentration than the MS. The ingredient and chemical composition of the experimental diets are given in Table 2. The differences observed due to the partial replacement of GS with MS were, on average, lower dietary concentrations of CP (157 vs. 136 g/kg DM), NDF (363 vs. 340 g/kg DM), PVB (26.7 vs. 6.0 g/kg DM), and pdNDF (303 vs. 273 g/kg DM). Adding RSO to the diets increased the concentration of crude fat on average by 39.3 g/kg DM, and consequently the ME by 0.9 MJ/kg DM.

### 3.3. Intake, Milk Production and Efficiency

There were no significant interactive effects of forage source and RSO supplementation on production parameters (*p* ≥ 0.29), except for intake of PVB and milk urea (MU) (*p* < 0.01), which were highest for cows fed the GS diet (Table 3). Partial replacement of GS with MS decreased (*p* < 0.01) total DMI by 1.0 kg/d and intake of silage by 0.4 kg/d. Similarly, the intake of OM, CP, NDF, pdNDF, ME, and MP was lower (*p* ≤ 0.05) when GS was replaced with MS in the experimental diets. The MY and ECM yield decreased (*p* ≤ 0.01) by 2.7 and 2.5 kg/d, respectively, and yields of fat, protein, and lactose decreased by 102, 85, and 116 g/d, respectively, when MS was included in the diets. Nitrogen efficiency improved (*p* ≤ 0.01) in cows fed the diets with MS compared with cows fed the diets with GS as the sole forage. Supplementing diets with RSO decreased (*p* < 0.01) DMI and silage intake by 1.9 and 1.5 kg/d and reduced (*p* ≤ 0.01) the intake of OM, CP, NDF, iNDF, pdNDF, and MP. Cows fed diets with RSO increased (*p* ≤ 0.05) their MY by 0.8 kg/d, but reduced their yield of ECM by 2.6 kg/d. Adding RSO to the experimental diets decreased (*p* < 0.01) milk fat concentration and yield of fat by 7.8 g/kg and 187 g/d, respectively, and protein concentration and yield by 2.8 g/kg and 48 g/d, respectively, while the concentration and yield of lactose increased (*p* < 0.01) by 0.1 g/kg and 116 g/d, respectively. 

### 3.4. Apparent Digestibility of Nutrients

The effect of replacing GS with MS, with or without supplementation with RSO, on nutrient digestibility is shown in Table 4. Feeding the MS diets decreased (*p* < 0.01) the apparent digestibility of DM, OM, CP, NDF, and pdNDF, by 16, 19.5, 160, 62.5, and 57.5 g/kg, respectively. Similar results were observed for RSO supplementation, which decreased (*p* < 0.01) the digestibility of DM, OM, NDF, and pdNDF by 27.0, 25.5, 47.5, and 62.5 g/kg, respectively, compared with cows fed diets without RSO supplementation.

### 3.5. Gas Emissions

Inclusion of residual CH_4_ as covariate was significant for all measures of CH_4_ emissions, daily emissions, and yield of CO_2_, CH_4_/CO_2_ ratio, and O_2_ consumption (*p* ≤ 0.03). However, if not included in the model it did not change the magnitude of any of the given gas emission traits (Table 5). Partial replacement of GS with MS increased (*p* ≤ 0.01) CH_4_ and CO_2_ intensity by 0.8 and 15.5 g/kg of ECM, respectively. Cows fed diets with MS decreased (*p* < 0.01) their daily CO_2_ emissions by 609 g/d and O_2_ consumption by 448 g/d. Diet supplementation with RSO reduced (*p* ≤ 0.01) daily CH_4_ emissions, yield, and intensity by 100 g/d, 3.0 g/kg of DM, and 2.1 g/kg of ECM, respectively. It also decreased daily CO_2_ emissions g/d, CH_4_/CO_2_ ratio, O_2_ consumption g/d and RQ by 974 g/d, 6.2, 451 g/d, and 0.02, respectively, in comparison with cows fed diets not supplemented with RSO.

### 3.6. Rumen Microbiota

Pearson correlation coefficients (PCC) ≥ 0.83 were found for all five mocks included in this study as a proxy to validate the accuracy of the sequencing process. Archaea represented between 1.58% and 2.09% of the reads obtained, and the ratio of archaeal over bacterial reads was not significantly different between diets (*p* ≥ 0.14). The relative abundance of rumen archaea is presented in Figure 2. In cows fed diets supplemented with RSO, Methanosphaera relative abundance was increased by 0.58%. A number of rumen bacterial genera were identified. The 20 most abundant taxa with average relative abundances at family and genus level >1% are presented in Figure 3. The diets containing MS decreased (*p* ≤ 0.04) the relative abundance of *Ruminococcaceae*/*Ruminococcus_1*, *Atopobiaceae*/*Olsenella*, and *Veillonellaceae*/*Selenomonas_1* by 0.54, 0.41, and 0.31 points, respectively. Adding RSO to the diet also lowered (*p ≤* 0.05) the abundance of *Bifidobacteriaceae*/*Bifidobacterium* by 3 points, *Lachnospiraceae*/uncultured by 1.0 point, *Atopobiaceae*/*Olsenella* by 0.6 points, *Prevotellaceae*/UCG001 by 1.2 points, and *Veillonellaceae*/*Selenomonas_1* by 0.3 points. However, adding RSO to the dairy cow diets increased (*p* ≤ 0.04) the relative abundance of *Succinivibrionaceae*/UCG002 by 2.8 points, *Succinivibrionaceae*/UCG001 by 4.7 points, and *Succinivibrionaceae/Succinivibrio* by 2.3 points. Further, the total copy number of protozoal 18S r RNA gene copies per mL of rumen fluid was reduced (*p* < 0.01) from 4.45 to 2.06 × 10^5^ when RSO was added to the dairy cow diets.

## 4. Discussion

Replacing GS with MS is suggested to have a CH_4_-mitigating effect by causing a shift in rumen fermentation promoting increased propionate production, while the addition of oilseeds to the diet of ruminants can also shift the VFA profile towards more propionate and less acetate [40,41,42]. However, other underlying mechanisms have primarily been credited with the mitigation of CH_4_ emissions arising from dietary oil supplementation. Non-fermentable fatty acids decrease the extent of fermentation in the rumen, leaving a smaller amount of H_2_ available for methanogenesis. Alternatively, a direct inhibitory effect of unsaturated fatty acids on methanogens has been suggested, with dietary fat suppressing the function of ruminal protozoa and fibre-digesting microbes, biohydrogenation of unsaturated fatty acids capturing H_2_ and acting as an alternative H_2_ sink, or dietary fat simply mitigating CH_4_ emissions as a consequence of depressed DMI [43]. Benchaar et al. [42] observed a greater CH_4_ emissions-mitigating effect of linseed oil supplementation with a diet based on MS rather than red clover silage, but the performance of cows fed the MS-based diet was lower than that of cows fed red clover silage-based diets.

Thus, potential CH_4_ emission-mitigating mechanisms of dietary supplementation with unsaturated oil may be modified by the effect of the basal forage type on rumen fermentation. Prior to the present study, the effect on dairy cows of unsaturated oil supplementation of diets containing different forage sources had not been well established and it was not known whether the mitigating effect of diet on CH_4_ emissions is of equal magnitude regarding individual cow CH_4_-emitting characteristics.

### 4.1. Intake, Milk Production and Apparent Digestibility of Nutrients

Total DM and nutrient intake decreased when GS was replaced with MS in the diets in this experiment. Replacement of GS with MS resulted in lower yield of milk and milk components, most likely as a result of the lower DMI. However, Brask et al. [44] and Arndt et al. [45] observed no difference in DMI or milk yield when feeding GS of different maturity compared with MS, or when changing the ratio of alfalfa silage to MS in the diet of dairy cows. In contrast to findings by Brask et al. [44] and Arndt et al. [45] and in this study, Hart et al. [46] observed increased DMI and milk production when replacing GS with MS in diets fed to dairy cows. Law et al. [47] found that increasing the dietary protein content from 14.4 to 17.3 % improved milk production for cows in early lactation, but not for cows in later lactation. The cows in the study by Brask et al. [44] were on average in a more advanced stage of lactation and yielded less milk than the cows used in our experiment. It is likely that the cows in the present study consumed more and yielded more milk due to average higher dietary CP when fed GS diets compared with diets with MS.

Maize silage is not equivalent to GS from a nutritional perspective, due to its lower CP, NDF and pdNDF and, particularly, its high starch content. As suggested by Gadeken and Casper [48], particularly decreased dietary pdNDF content could be the reason for lower intake of DM. Daily intake of starch increased on average from 3.7 to 5.2 kg when GS was replaced with MS in the diets in this study (results not presented). The compositional differences between GS and MS specifically resulted in a lower intake of CP and pdNDF, and consequently the digestibility of CP and pdNDF decreased when MS replaced GS in the experimental diets. This is in agreement with Brask et al. [44] and van Gastelen et al. [7], who found that the digestibility of starch increased, and the digestibility of NDF decreased, with an increased proportion of MS in diets fed to dairy cows. In studies where MS has been found to increase DMI and milk yield, the lower intake and digestibility of CP and NDF is likely compensated for by increased intake and digestibility of starch, resulting in comparatively greater total digestibility of organic matter. We did not analyse starch in all dietary ingredients or in faecal samples and, moreover, the replacement rate of GS by MS was moderate compared with the study by van Gastelen et al. [7], who observed a linear increase in intake with increased MS proportion in the diet. Khan et al. [49] reviewed the nutritive value and milk yield response of the inclusion of MS in GS-based diets and concluded that a variation in the quality of MS, and of GS, will affect the optimum inclusion level of MS in diets for dairy cows.

Milk urea concentration was lower for the GSMS diets compared with the GS diets. In line with the lower dietary CP concentration, N efficiency increased when MS was fed to the cows, which has been reported as the primary nutritional factor determining N efficiency [50]. The lower N intake and the decrease in MU observed for the MS and RSO-supplemented diets suggest greater N retention [21,51]. In general, the experimental diets provided enough CP, except for the GSMSO that showed a slightly lower MU concentration compared to the adequate range of 2.8–4.2 mmol/L as suggested by Ishler [52].

Dietary supplementation with RSO further decreased intake of DM and nutrients, and yield of ECM. Supplementation of dairy cow diets with plant oils has previously been found to increase dietary ME concentration and potentially increase milk yield, but a lack of fermentable energy substrates in the rumen of cows fed these diets can negatively affect milk fat and protein synthesis, and subsequently ECM yield [42,53]. Benchaar et al. [42] reported a decrease in ECM yield of 2 kg/d, i.e., close to that observed in the present study, on including linseed oil at 40 g/kg DM in dairy cow diets. In the present study, supplying 41 g/kg of RSO significantly decreased the digestibility of DM, OM, NDF, and pdNDF, which are the most commonly observed effects when unsaturated fats are fed to ruminants [54,55,56]. However, Bayat et al. [8] found no effect on the apparent digestibility of nutrients when RSO at 50 g/kg of DM was included in diets fed to dairy cows. Inconsistencies between studies can be attributed to differences in fibre composition of the basal diet and in the levels and physical forms of dietary fatty acids [57]. Further, the presence of unsaturated lipids is damaging to some bacteria and ciliate protozoa, reducing the fibrolytic bacterial activity [58,59] and also modifying the rumen microbiota, as observed in our experiment.

### 4.2. Gas Emissions and Effect of Individual Cow Pre-Trial Measured CH_4_ Emissions

Partial replacement of GS with MS did not alter daily emissions and overall yield of CH_4_. Only the CH_4_ intensity was slightly increased, due to the decreased ECM yield when the cows were fed MS. A CH_4_ emissions-mitigating effect has been reported when MS is used as the sole forage source compared with red clover silage [42] and when more than 70% of GS is replaced with MS [7,60]. The factors influencing CH_4_ enteric production are primarily total DMI and diet OM digestibility and dietary fat and fibre content [10]. The observed decrease in intake when GS was replaced with MS was relatively small in this study and the decreased digestibility of NDF and CP was likely compensated for by the increased digestibility of the maize starch. Further, the relatively moderate proportion of MS in the experimental diets was probably not sufficient to modify the rumen fermentation pattern. The decreases seen in total CO_2_ production, CO_2_ intensity and total O_2_ consumed were in agreement with a decrease in intake and ECM production when GS was replaced with MS in the experimental diets.

A CH_4_ emissions-mitigating effect of oilseed supplementation in ruminant diets has been reported in several studies [53,61,62]. In the present study, feeding RSO at 41 g/kg of DM decreased daily CH_4_ emissions by 22.5%. Bayat et al. [8] reported a decrease in CH_4_ emissions of similar magnitude (22.6%) when the diet of dairy cows was supplemented with RSO at 50 g/kg of DM. Supplementation with RSO decreased CH_4_ emissions in the present study, indicating that mechanisms in the rumen caused the reduction, rather than solely a depressed intake, supporting previous findings [5]. The CH_4_/CO_2_ ratio describes the proportion of unmetabolised C relative to excreted CO_2_ and the low ratio observed when RSO was added to the diets indicates inefficiency in microbial fermentation of the feed [63]. Supplementation with RSO also resulted in lower RQ than in the cows not fed a diet supplemented with oil. These results indicate that feeding unsaturated fat to dairy cows slightly affects energy metabolism, since fat generally lowers the RQ.

Animal factors also play a significant role in enteric CH_4_ emissions [64,65]. Studies on sheep have shown that variation in ruminal digesta retention time affects CH_4_ emissions, with high CH_4_ emitters having a larger rumen volume and digesta pools than low emitters [66,67,68]. Other studies have shown that the host animal controls the archaea population in the rumen [69,70]. However, Cabezas-Garcia et al. [71] observed dietary variations in molar proportion of VFA and found that the effect of VFA on CH_4_ production was much greater than the corresponding effect of variations in animals. This suggests that rumen fermentation patterns are more strongly associated with differences in fermented substrates deriving from the diets than with differences in rumen microbiota between cows. This supports the finding in this study of no variation in dietary mitigation of CH_4_ emissions when not applying or applying a covariate correction of individual cow CH_4_-emission characteristics in the statistical model.

### 4.3. Rumen Microbiota

Of the 20 most abundant bacterial taxa at the family/genus level, only three were marginally influenced by the MS diets and, to our knowledge, these genera are not strongly associated with methanogenesis in the rumen. Poulsen et al. [72] added RSO at 33 g/kg of DM in an in vitro study and observed a reduction in CH_4_ production related to depletion in the relative abundance of *Thermoplasmata* (*Methanomassiliicoccaceae*) and an increase in the relative abundance of both *Methanosphaera* and *Methanobrevibacter*. Other studies testing lipid inclusion in ruminant diets have also reported mitigation of CH_4_ emissions related to increased *Methanosphaera* and *Methanobrevibacter* abundance [8,73,74]. In the present study, *Methanobrevibacter* was identified as the most abundant archaea (~95%)*,* and *Methanosphaera* levels increased significantly when RSO was fed to the cows.

The bacterial community in the rumen cooperates with archaea to produce enteric CH_4_. At the family level, *Prevotellaceae* and *Succinivibrionaceae* were the most abundant bacterial taxa observed in the dairy cow rumen under our experimental conditions. Rapeseed oil supplementation affected eight bacterial taxa at the family/genus level. Among these, the relative abundance of the *Bifidobacteriaceae* decreased, while that of *Succinivibrionaceae* increased substantially. It is well known that members of the *Bifidobacteriaceae* are associated with greater lactic and acetic acid production, instead of production of reduced substances such as propionate [75]. Consequently, more H_2_ is available for CH_4_ molecule formation by the methanogenic archaea in the rumen [76], which explains the greater CH_4_ emissions observed here for the diets without RSO supplementation. On the other hand, an increase in *Succinivibrionaceae* relative abundance in the rumen is related to lower CH_4_ emissions in ruminants [77,78]. These bacteria incorporate H_2_ to produce succinate, which is further metabolised to propionate by other ruminal microorganisms [79], and thus less H_2_ is available in the rumen and less CH_4_ is produced.

Total count of ciliate protozoal 18S rRNA gene copies was also significantly decreased by RSO supplementation of the dairy cow diets. A reduction in protozoa numbers in ruminants due to oil supplementation has been reported previously for sunflower oil [80], maize oil [81], soybean oil [82] and linseed oil [83]. Protozoa establish symbiotic associations with prokaryotes in the rumen, among which their association with archaea plays a key role in methanogenesis. Thus, a reduction in protozoa (and therefore their symbiotic methanogens) can be expected to be correlated with a reduction in CH_4_ emissions [84,85]. Furthermore, *Methanobrevibacter* has been proposed as a methanogen predominantly associated with protozoa [84,86]. Interestingly, our results are in line with both statements, since the dietary treatments yielding less CH_4_ resulted in fewer protozoa in the rumen and a lower proportion of *Methanobrevibacter*, suggesting that it is potentially a protozoal symbiont.

## 5. Conclusions

Replacing GS with MS in diets fed to dairy cows negatively affected nutrient intake, nutrient digestibility, and milk production. Supplementation with RSO at 41 g/kg dietary DM impaired animal performance and caused modifications in the rumen microbiota, but effectively reduced CH_4_ emissions by 22.5%. The pre-trial residual CH_4_ emissions level of the individual cows did not affect the magnitude of the mitigating effect of the diets on CH_4_ emissions, indicating that the effect of CH_4_-emitting phenotype might be negligible in comparison with the mitigating effect of specific dietary ingredients on CH_4_ production in dairy cows.

## Figures and Tables

**Figure 1 animals-11-02597-f001:**
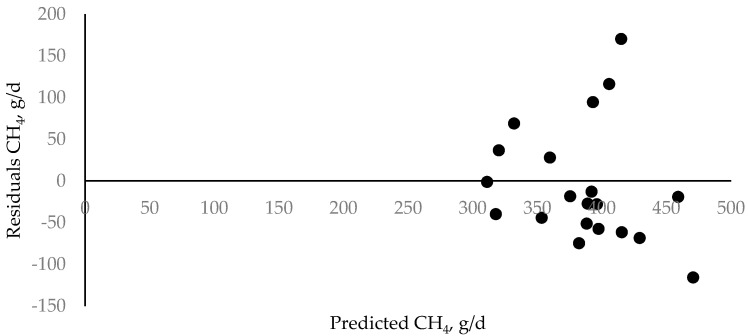
Relationship between predicted methane (CH_4_) emissions and residual CH_4_ emissions (CH_4_ observed − CH_4_ predicted) (*n* = 20).

**Figure 2 animals-11-02597-f002:**
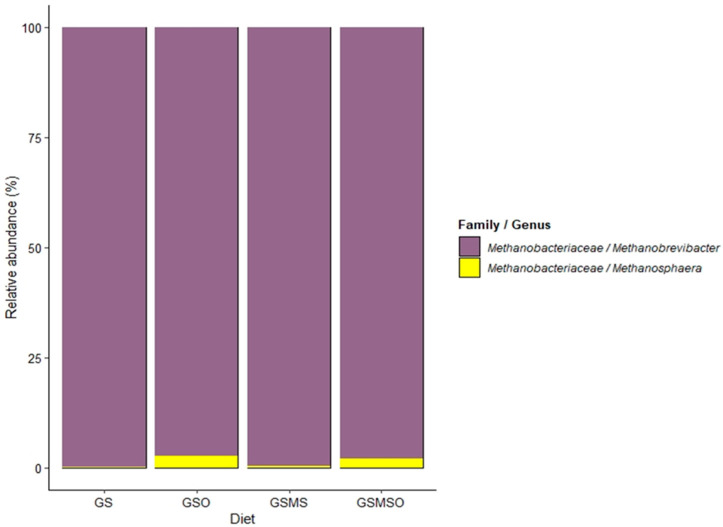
Archaea family and genus composition in rumen fluid, shown as mean percentage relative abundance for each experimental diet: GS—grass silage; GSO—grass silage with rapeseed oil supplementation; GSMS—grass silage plus maize silage; GSMSO—grass silage plus maize silage with rapeseed oil supplementation.

**Figure 3 animals-11-02597-f003:**
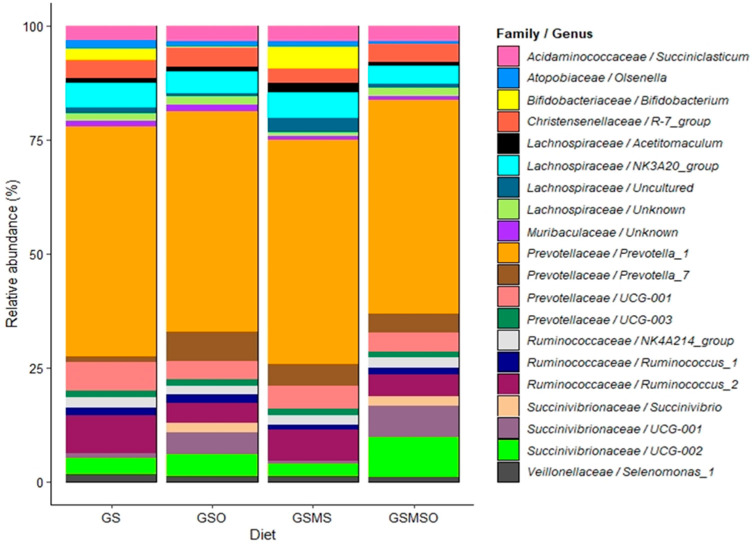
Bacteria family and genus composition in rumen fluid, shown as mean percentage relative abundance for each experimental diet: GS—grass silage; GSO—grass silage with rapeseed oil supplementation; GSMS—grass silage plus maize silage; GSMSO—grass silage plus maize silage with rapeseed oil supplementation.

**Table 1 animals-11-02597-t001:** Chemical composition and nutritional values of the dietary ingredients used in experimental diets fed to dairy cows (g/kg of DM unless otherwise stated).

Item ^1^	Dietary Ingredient				
	Grass Silage	Maize Silage	Crimped Barley	Rapeseed Meal ^2^	Concentrate ^3^
Dry matter, g/kg	294	432	590	870	883
Chemical composition					
Organic matter	921	966	966	916	922
Crude protein	142	65.3	142	371	222
Neutral detergent fibre (NDF)	529	438	161	240	254
Indigestible NDF (iNDF)	66.6	90.0	45.2	94.5	64.2
pdNDF	458	348	113	146	190
Crude fat	35.0	31.7	19.0	86.8	60.0
Starch	NA ^4^	320	503	16.0	357
Fermentation quality					
pH	3.75	3.89	-	-	-
Ammonia-N, g/kg of N	47.1	106	-	-	-
Lactic acid	99.0	51.7	-	-	-
Acetic acid	21.7	19.3	-	-	-
Butyric acid	0.62	0.38	-	-	-
Nutritional values					
ME, MJ/kg of DM	11.5	11.3	13.2	11.4	13.3
MP, g/kg of DM	84	81	90	169	112
PVB, g/kg of DM	35	−38	−20	154	46

^1^ pdNDF—potentially digestible NDF (NDF-iNDF); ME—metabolisable energy and PVB—protein balance in the rumen, both calculated based on coefficients from feed tables [20]; MP—metabolisable protein calculated according to [21]; ^2^ ExPro-00SF (AarhusKarlshamn Ltd., Malmö, Sweden); ^3^ Commercial concentrate used in GreenFeed (Komplett Amin 220; Lantmännen Lantbruk AB, Malmö, Sweden); ^4^ NA—not analysed.

**Table 2 animals-11-02597-t002:** Ingredient and chemical composition, and nutritional values of experimental diets fed to dairy cows (g/kg of DM); mean ± (SD).

Item ^1^	Diet ^2^			
	GS	GSO	GSMS	GSMSO
Ingredient composition (*n* = 20)				
Grass silage	560 (13.1)	530 (13.7)	276 (11.5)	283 (11.9)
Maize silage	0	0	290 (8.5)	258 (6.7)
Crimped barley	330 (10.3)	318 (9.3)	328 (5.9)	314 (5.20)
Rapeseed meal	90 (2.8)	90 (2.7)	90 (1.6)	90 (1.3)
Rapeseed oil	0	42 (4.2)	0	39 (2.8)
Mineral mixture	20 (0.46)	20 (0.45)	16 (0.27)	16 (0.24)
Chemical composition (*n =* 4)				
Organic matter	922 (3.3)	924 (2.9)	935 (2.1)	935 (1.6)
Crude protein	160 (9.0)	154 (3.1)	138 (9.3)	133 (4.5)
Neutral detergent fibre (NDF)	371 (10.3)	355 (5.4)	348 (14.4)	332 (9.7)
Indigestible NDF (iNDF)	61.9 (3.8)	59.4 (2.0)	68.8 (3.8)	66.1 (2.2)
pdNDF	309 (10)	296 (8.7)	279 (9.1)	266 (6.7)
Crude fat	35.1 (0.57)	75.3 (4.1)	34.6 (0.78)	72.9 (2.6)
Nutritional values (*n =* 4)				
ME, MJ/kg of DM	11.9 (0.04)	12.8 (0.08)	11.8 (0.03)	12.6 (0.05)
MP, g/kg of DM	92.5 (0.49)	88.7 (0.71)	92.0 (0.35)	88.6 (0.41)
PVB, g/kg of DM	26.9 (0.30)	26.4 (0.41)	5.9 (0.98)	6.1 (0.65)

^1^ Rapeseed meal is ExPro-00SF (AAK Sweden AB, Karlshamn, Sweden); Mineral mixture—Mixa Optimal (Lantmännen Lantbruk AB, Malmö, Sweden); pdNDF—potentially digestible NDF (NDF-iNDF); ME—metabolisable energy and PVB—protein balance in the rumen, both calculated based on coefficients from feed tables [20]; MP—metabolisable protein calculated according to Spörndly [21]; ^2^ GS—grass silage; GSO—grass silage with rapeseed oil supplementation; GSMS—grass silage plus maize silage; GSMSO—grass silage plus maize silage with rapeseed oil supplementation.

**Table 3 animals-11-02597-t003:** Intake and production data for cows fed the experimental diets (*n* = 20).

Item ^1^	Diet ^2^				SEM	*p*-Value ^3^	
	GS	GSO	GSMS	GSMSO		Forage	Oil
Intake, kg/d							
Total DM	21.6	19.9	20.7	18.8	0.27	<0.01	<0.01
Silage DM	11.4	10.0	11.1	9.4	0.14	<0.01	<0.01
Organic matter	19.9	18.4	19.3	17.6	0.25	<0.01	<0.01
Crude protein	3.6	3.2	3.0	2.6	0.04	<0.01	<0.01
Neutral detergent fibre (NDF)	7.6	6.8	6.8	6.0	0.19	<0.01	<0.01
Indigestible NDF (iNDF)	1.2	1.1	1.3	1.2	0.04	0.18	<0.01
pdNDF	6.4	5.7	5.6	4.8	0.17	<0.01	<0.01
ME, MJ/d	256	254	246	235	3.7	<0.01	0.08
MP, kg/d	2.0	1.78	1.92	1.60	0.035	0.05	<0.01
PVB, kg/d	0.59	0.54	0.16	0.16	0.02	<0.01	<0.01
Milk yield, kg/d	31.5	32.4	28.8	29.5	0.39	<0.01	0.05
ECM yield, kg/d	34.3	32.1	31.8	29.0	0.57	<0.01	<0.01
Milk composition, g/kg							
Fat	46.2	39.7	47.6	38.5	0.80	0.84	<0.01
Protein	36.7	33.7	36.6	34.0	0.36	0.93	<0.01
Lactose	45.2	46.1	45.0	46.2	0.15	0.72	<0.01
MU, mmol/L	3.99	2.91	3.19	2.74	0.124	<0.01	<0.01
Composition yield, g/d							
Fat	1428	1283	1368	1138	31.5	<0.01	<0.01
Protein	1135	1091	1054	1001	16.7	<0.01	<0.01
Lactose	1399	1491	1295	1362	26.3	<0.01	<0.01
Feed efficiency, kg/kg	1.59	1.63	1.56	1.56	0.056	0.09	0.51
N efficiency, g/kg	313	339	352	377	6.3	<0.01	<0.01

^1^ pdNDF—potentially digestible NDF (NDF-iNDF); ME—metabolisable energy, MP—metabolisable protein, and PVB—protein balance in the rumen calculated based on coefficients from feed tables [20] and according to Spörndly [21]; ECM—energy-corrected milk calculated according to Sjaunja et al. [37]; MU—milk urea. Feed efficiency = ECM/total DM intake; N (nitrogen) efficiency = milk N/ N intake; ^2^ GS—grass silage; GSO—grass silage with rapeseed oil supplementation; GSMS—grass silage plus maize silage; GSMSO—grass silage plus maize silage with rapeseed oil supplementation; ^3^ Probability of significance of the effect of forage type and rapeseed oil, and of the interaction between forage × oil; the interaction was not significant for any item (*p* ≥ 0.29) except PVB and MU (*p* < 0.01).

**Table 4 animals-11-02597-t004:** Digestibility of dietary chemical components in cows in the experiment (g/kg; *n* = 20).

Item ^1^	Diet ^2^				SEM	*p*-Value ^3^	
	GS	GSO	GSMS	GSMSO		Forage	Oil
Dry matter	747	723	732	704	7.9	<0.01	<0.01
Organic matter	768	746	753	723	7.5	<0.01	<0.01
Crude protein	659	676	497	518	14.7	<0.01	0.15
Neutral detergent fibre (NDF)	626	584	570	516	11.2	<0.01	<0.01
pdNDF	745	694	699	625	14.2	<0.01	<0.01

^1^ pdNDF—potentially digestible NDF (NDF-indigestible NDF); ^2^ GS—grass silage; GSO—grass silage with rapeseed oil supplementation; GSMS—grass silage plus maize silage; GSMSO—grass silage plus maize silage with rapeseed oil supplementation; ^3^ Probability of significance of the effect of forage type and rapeseed oil, and of the interaction between forage × oil; the interaction was not significant for any item (*p* ≥ 0.35).

**Table 5 animals-11-02597-t005:** Methane (CH_4_), carbon dioxide (CO_2_) emissions and oxygen (O_2_) consumption for cows fed the experimental diets (*n* = 20).

Item ^1^	Diet ^2^				SEM	*p*-Value ^3^	
	GS	GSO	GSMS	GSMSO		Forage	Oil
CH_4_							
g/d	453	351	440	341	13.0	0.27	<0.01
g/kg of DMI	20.9	17.9	21.7	18.6	0.77	0.13	<0.01
g/kg of ECM	13.3	11.0	14.0	12.0	0.40	0.01	<0.01
CO_2_							
g/d	12590	11695	12060	11006	221.4	<0.01	<0.01
g/kg of DMI	585	594	594	593	15.7	0.66	0.70
g/kg of ECM	370	368	382	387	8.0	0.02	0.86
CH_4_/CO_2_, g/kg	35.8	29.9	36.6	30.1	0.70	0.14	<0.01
O_2,_ g/d	9117	8724	8727	8217	147.9	<0.01	<0.01
RQ	1.00	0.98	1.00	0.98	0.005	0.85	0.01

^1^ DMI—dry matter intake; ECM—energy-corrected milk; RQ—respiratory quotient (CO_2_ emitted/O_2_ consumed); ^2^ GS—grass silage; GSgrass silage with rapeseed oil supplementation; GSMS—grass silage plus maize silage; GSMSO—grass silage plus maize silage with rapeseed oil supplementation; ^3^ Probability of significance of the effect of forage type and rapeseed oil, and of the interaction between forage × oil; the interaction was not significant for any item (*p* ≥ 0.51).

## Data Availability

The raw sequence data for the microbial analysis that was generated for this study can be found in the European Nucleotide Archive (ENA) under accession number PRJEB43834 (archaea and bacteria) and AM158474.1 (protozoa).

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
