# Peer review of "Effect of a Low-Methane Diet on Performance and Microbiome in Lactating Dairy Cows Accounting for Individual Pre-Trial Methane Emissions"

_animals, 2021, doi:10.3390/ani11092597_

Round 1

Reviewer 1 Report

Reviewer comments:

Please see below some minor corrections in terms of grammar and punctuation.

Line 78: ‘would’ produce …..moderate supplementation with RSO ‘would’ further

Line 96: Prior not previously

Line 96: was carried out not ‘set’

Line 98: days in milk vs DIM?

Line 99: delete  pre-trial at the end of the sentence

Line 99-103: Please include a space after : and a full stop after DM at the end of the sentence.

Line 96-106: Please include a reference here to support this methodology and the duration of the 7 day pre-trial to establish individual cow characteristics of methane emissions.

What was the range of individual cow methane emissions?

Line 109: ad libitum should be in italics

Line 115: experimental periods lasting 28 d and totalizing 112 d of experiment – rephrase ‘experimental periods lasting 28 d each with a total experiment duration of 112 d’

Line 115: All data were recorded and samples were collected during the last 14 d of each experimental period.

Line 127: any metrics around the amount of red clover in the sward?  The use of the word some suggested to me that it was not at the desired level?

Line 321: Sentence structure – brackets after mean and SD values

Line 421: add a space between CH4 and emissions

Author Response

Answer to reviewers’ comments on Effect of a low-methane diet on performance and microbiome in lactating dairy cows accounting for individual pre-trial methane emissions by Chagas et al.

AU: We responded to comments in the usual line-by-line manner. When changes in the manuscript follow from any comment, it will be seen at the end of the answer by giving the line numbers of where the changes are to be found in the revised manuscript. Answers are highlighted in yellow (file attached) and the changes in the revised manuscript are tracked.

Reviewer 2 Report

Authors addressed comments/concerns from the previous version.

Author Response

(The authors gave the same response as above.)

Reviewer 3 Report

Thanks for your answers and clarifications and congratulations for your work.

Regarding my remark on "direct inhibition of fat synthesis by fat in the rumen", please, see bellow the original work on the subject that was followed by many other on this topic:

Griinari, J. M., D. A. Dwyer, M. A. McGuire, D. E. Bauman, D. L. Palmquist, and K. V. V. Nurmela. 1998. Trans-octadecenoic acids and milk fat depression in lactating dairy cows. J. Dairy Sci. 81: 1251-1261

Feel free to incorporate or not this on your manuscript.

Author Response

Answer to reviewers’ comments on Effect of a low-methane diet on performance and microbiome in lactating dairy cows accounting for individual pre-trial methane emissions by Chagas et al.

AU: We responded to comments in the usual line-by-line manner. When changes in the manuscript follow from any comment, it will be seen at the end of the answer by giving the line numbers of where the changes are to be found in the revised manuscript. Answers are highlighted in yellow (file attached) and the changes in the revised manuscript are tracked.

This manuscript is a resubmission of an earlier submission. The following is a list of the peer review reports and author responses from that submission.

Round 1

Reviewer 1 Report

Reviewer comments:

Overall this is a very well-structured paper that tests a good hypothesis and explains clearly the findings and outcomes.

Please see below some minor corrections:

Line 3: Lactation not lactating

Line 12-14: Suggestion to restructure: It is clearer to understand if you explain the effect of first before the effect on: ‘This study investigated the effects of replacing grass silage with maize silage, with or without rapeseed oil supplementation, on methane emissions  of dairy cows of a low or high methane-emitting phenotype.

Line 14 & 18: The aim of the simple summary is for a non-specialist to understand the work. Therefore an alternative word for phenotype should be included.

Line 20-22: restructure: ‘This study examined the of partly replacing grass silage (GS) in the diet of lactating dairy cows with maize 21 silage (MS), with or without rapeseed oil (RSO) supplementation on methane (CH4) emissions, production performance and 20 rumen microbiome’

Line 24 & 95 and elsewhere: brackets should follow the statement not come before it

Line 94: consistency with numbers or works 15 vs five. Please correct

Line 107: CH4 production was measured over 7 day to determine the CH4 emitting phenotype of each cow. Why 7 days? Please provide some further information here and a reference to support that 7 days is a sufficient duration of time to categorise if an animal is a low or high emitter of CH4.

Line 112-119: This section could be structured better for the reader to follow. Please restructure. It is difficult to follow since ‘respectively’ and the ‘units’ come before the information.

Table 1:

Why is the starch content not reported for the concentrate and barley? I would suggest including this data for completeness 

Consistency with the number of decimal places and the structure. Some sections are lined up based on decimal places but the chemical composition section is not.

Organic matter and Ammonia-N methods are not described in the M&Ms section - please add

Line 145, 147: Milk yield was abbreviated previously?

Line 211: urea – Table 3 uses MU consistency required.

Line 326: NH3 is not abbreviated previously

Table 2:

Please indicated what the numbers in brackets under the diets refer to. E.g. 560 (13.1) If this is referring to the ingredient composition on an individual animal basis then the diet allocations are much higher than the Total DMI in Table 3. Please explain.

The starch content of the experimental diets was not analyzed?

Author Response

The answers to your comments are highlighted in the attached file.

Reviewer 2 Report

General comments:

This manuscript evaluates the inclusion of maize silage as a replacement of grass silage and the inclusion of rapeseed oil in dairy cow diets.  While residual methane is included as a covariate to account for differences between animals during a common feeding period, evaluation of whether low or high methane-emitting phenotypes are impacted by diet was not directly evaluated. If low and high methane emitting phenotype was actually part of the treatment assignment, then it is possible that interpretation of the results could be quite different.  There is a valid experiment described here, but it is really about whether the replacement of grass silage and rapeseed oil affects performance and the microbiome.  The title should be revised to reflect the treatments evaluated.  Consider removing “of low or high methane-emitting phenotype” from the title because that was not directly evaluated in this experiment. Also, how long did the study last?  This is not clear.  If there are 4 period of 28 days, then the length of the study would be 112 days. If the average DIM at the start of the study was 71 days, most of the cows were in mid-lactation during the extent of the study.  Revise “early lactation” to mid-lactation.

Experimental design and statistics:  Line 94-95 states that there were 15 multiparous and 5 primiparous cows used in this experiment.  Line 101-104 describes the experimental design as a 4 x 4 Latin square design.  The statistical model is described on Line 291. If cows were blocked by parity and milk yield (as described on Lines 101-102), there should be 5 blocks (or squares) of 4 cows each to assign cows to.  If there are 5 primiparous cows, then how was parity handled?  Were 4 primiparous cows in one block and the 5th primparous in with multiparous cows?  Or was there one primiparous cow in each of the 5 blocks (or squares)?  This is not clear.  It currently reads that the all the primiparous cows are blocked together, but then there is an extra primiparous cow. How were cows blocked by milk production? Highest producers all grouped together?  Typically in Latin square designs, interactions of the square and diet can be evaluated in the statistical model.  That is not described on Line 291. Any reason why this wasn’t evaluated?

Wouldn’t cows be assigned to a square rather than a block in reference to Line 102-104 describing the study as a “replicated 4 x 4 Latin square design”.   How many periods were included in this evaluation?  Please state this in the materials and methods section. Currently it in not clear how many experimental units are in the experiment.  Perhaps there were fewer than 4 periods?  If there were 4 periods, then all 20 cows were on the Greenfeed system to measure methane production 5 times [once during the pre-trial period for 7 days (line 107), and 4 more times during d 15 to 28 d of each period (line 157)].  How often did cows visit the Greenfeed system during that one week or 2 week recording period?  There is no mention of this in the materials and methods.

Regarding the statistical model on Line 291 (and 299), there is no mention of evaluating the interaction of forage and oil inclusion.  Since this is described as a 2 x 2 factorial arrangement of treatments (Line 111), why isn’t dietary factor 1 (forage), dietary factor 2 (oil inclusion) and the interaction of these 2 factors included in the statistical model?

In reference to the inclusion of the covariate in model 2 (line 299).  Is the residual methane as a covariate significant?  Can it be removed from the model without affecting the interpretation of the data?  Conclusions imply that the inclusion of the pre-trial residual methane emission did not affect the magnitude of the effect of diets on methane emissions, therefore, it’s possible that it should not be included in the model at all.

Specific comments:

Figure 1: If n=20 (one for each cow), why are there 24 circles on this graph?

Lines 487-489:  What specifically is the reason for the observed lower DMI? This is not clear.  What specifically is suggested by Gadeken and Casper?

Lines 489-491:  Why not include starch composition and starch intake in Table 2 and 3?  Seems relevant to the discussion.

Lines 503-505:  Is Khan et al. (2016) the correct reference?  This is a review on transitioning from milk to solid feed in dairy calves.  How is this relevant to milk yield response of the inclusion of maize silage in a grass silage-based diet? 

Line 557-560:  Is reference [69] used in the paper? Overall there is a large number of references cited.  Are all of them relevant?  Consider reducing the number of references if they are not relevant. 

Author Response

(The authors gave the same response as above.)

Reviewer 3 Report

GENERAL COMMENTS

The manuscript contains interesting information of a well conceived and performed experiment, as well as with excellent readability.

Two important points should be clarified: (1) The fact of methane residual being used only as a co-variate would not allow comparisons between low and high emitting phenotype. It can be noticed in item “4.2 Gas emissions and effect of phenotype” that only literature data was used to support the effect of low and high emitting phenotype. Therefore, it was difficult to understand any consideration on the effect of low and high emitting phenotype like in the last paragraph of the conclusion.; (2) It was remarkable that no interaction was found for almost all variables (exceptions were PVB and MU). According to the usual practice, one would expect that only the average of the main effects should be presented for the data with no significant interaction.

Other comments and suggestions are bellow:

TITLE: Please, consider deleting the last part, after cows (unless proving this effect can be considered).

SIMPLE SUMMARY:

Line 16: Replace “but” by “due to”

ABSTRACT:

No suggestions, except by omitting this phrase “Cow CH4-emitting phenotype had a significant influence on gas emissions, but did not alter the magnitude of CH4 emissions” as this comparison was not addressed by the experiment.

 INTRODUCTION

Line 49: Reference 4 is from 2003 and so the 60% is not related the present time, as someone not checking the reference would assume.  Another point to consider is how advances of almost two decades in the production process and the inclusion of mitigating actions may have decreased the rate that was considered in a paper from the beginning of the century. It would be advisable to state in the text about this question of time or use a more recent citation.

MATERIAL AND METHODS

As one central point is on nutritional quality of GS and MS, therefore more information on both would be interesting. More info on MS: what was the cultivar? What was the harvest decision based on?  Specially for GS and crimped barley, one good info would be good for how many days they had at harvest

Line 100: How many blocks were created? It is impossible to know because the only clue is that cows were blocked by parity, but we have only information about the number of primiparous and multiparous cows, with no other detail and probably there were more than just those two blocks.

Lines 105-106:  The “TMR of 54:32:70 grass silage:crimped barley:commercial concentrate mixture” would equal 35:21:45 summing 100? Is that as fed or dry matter? Please, consider using the dry matter and 100 base mentioning it is dry matter.

Lines 109-110: I would suggest a section devoted exclusively to the pre-trial, as it quite important since in this phase the CH4 emission baseline for the cows was determined, as well as the MY used to block the animals. Perhaps, it should be the first section. The use of methane emission residuals was a good idea, but it is based in the premise that there is no interaction between diet type and animal. Perhaps mentioning this would be advisable.

RESULTS

Table 2: The values between brackets probably are a measure of dispersion but it should be stated in the title (or elsewhere) and the metric should be well defined (VC, SE, SEM, etc.)      

Line 338: How was It possible to measure the GS reduction separately? Perhaps more information on feeding in M&M would help.

Line 417:  Looking at Fig 2, the increase seems to be 58% and not 0,58%.  As a matter of fact, it seemed even greater, but it is difficult to figure out just l looking at the graph.

DISCUSSION

Lines 483-485: Higher CP levels of GS diet could have been the reason to higher MY but it would deserve further discussion since MP was almost the same for the diet with the inclusion of MS. Also, MUN was bellow the recommended range (lines 510-512) only for GSMSO (table 3).

Lines 487-488:  Were the differences in composition of MS high enough for, in the context of the whole diet to explain DMI reduction? Could the fact of being re-ensiled with a high DM have any consequence in MS quality? Although there is data on silage quality and it is OK, it is known that minor fermentation products that are not measured may influence DMI.

Line 525: The use of reference 57 is strange to discuss effects of fat on digestibility that are mostly on rumen.

Lines 513-528: The direct inhibition of fat synthesis by fat in the rumen was not considered? Please, consider including something about this possible effect.

Line 535: Paper 60 is maize silage replacing alfalfa silage, that is a legume, not a grass.

CONCLUSIONS

Lines 610-614: Based on what results was this last conclusion  based on?

Author Response

(The authors gave the same response as above.)
